# Efficient Thompson Sampling for Online Matrix-Factorization Recommendation

**Jaya Kawale, Hung Bui, Branislav Kveton**
Adobe Research
San Jose, CA
{kawale, hubui, kveton}@adobe.com

**Long Tran Thanh**
University of Southampton
Southampton, UK
ltt08r@ecs.soton.ac.uk

**Sanjay Chawla**
Qatar Computing Research Institute, Qatar
University of Sydney, Australia
sanjay.chawla@sydney.edu.au

## Abstract

Matrix factorization (MF) collaborative filtering is an effective and widely used method in recommendation systems. However, the problem of finding an optimal trade-off between exploration and exploitation (otherwise known as the bandit problem), a crucial problem in collaborative filtering from cold-start, has not been previously addressed. In this paper, we present a novel algorithm for online MF recommendation that automatically combines finding the most relevant items with exploring new or less-recommended items. Our approach, called *Particle Thompson sampling for MF* (PTS), is based on the general Thompson sampling framework, but augmented with a novel efficient *online* Bayesian probabilistic matrix factorization method based on the Rao-Blackwellized particle filter. Extensive experiments in collaborative filtering using several real-world datasets demonstrate that PTS significantly outperforms the current state-of-the-arts.

## 1 Introduction

Matrix factorization (MF) techniques have emerged as a powerful tool to perform collaborative filtering in large datasets [1]. These algorithms decompose a partially-observed matrix $R \in \mathbb{R}^{N \times M}$ into a product of two smaller matrices, $U \in \mathbb{R}^{N \times K}$ and $V \in \mathbb{R}^{M \times K}$, such that $R \approx UV^T$. A variety of MF-based methods have been proposed in the literature and have been successfully applied to various domains. Despite their promise, one of the challenges faced by these methods is recommending when a new user/item arrives in the system, also known as the problem of *cold-start*. Another challenge is recommending items in an online setting and quickly adapting to the user feedback as required by many real world applications including online advertising, serving personalized content, link prediction and product recommendations.

In this paper, we address these two challenges in the problem of online low-rank matrix completion by combining matrix completion with bandit algorithms. This setting was introduced in the previous work [2] but our work is the first satisfactory solution to this problem. In a bandit setting, we can model the problem as a repeated game where the environment chooses row $i$ of $R$ and the learning agent chooses column $j$. The $R_{ij}$ value is revealed and the goal (of the learning agent) is to minimize the cumulative regret with respect to the optimal solution, the highest entry in each row of $R$. The key design principle in a bandit setting is to balance between exploration and exploitation which solves the problem of cold start naturally. For example, in online advertising, exploration implies presenting new ads, about which little is known and observing subsequent feedback, while exploitation entails serving ads which are known to attract high click through rate.

While many solutions have been proposed for bandit problems, in the last five years or so, there has been a renewed interest in the use of Thompson sampling (TS) which was originally proposed in 1933 [3, 4]. In addition to having competitive empirical performance, TS is attractive due to its conceptual simplicity. An agent has to choose an action $a$ (column) from a set of available actions so as to maximize the reward $r$, but it does not know with certainty which action is optimal. Following TS, the agent will select $a$ with the probability that $a$ is the best action. Let $\theta$ denotes the unknown parameter governing reward structure, and $O_{1:t}$ the history of observations currently available to the agent. The agent chooses $a^* = a$ with probability

$$\int \mathbb{I}\left[\mathbb{E}\left[r|a, \theta\right] = \max_{a'} \mathbb{E}\left[r|a', \theta\right]\right] P(\theta|O_{1:t})d\theta$$

which can be implemented by simply sampling $\theta$ from the posterior $P(\theta|O_{1:t})$ and let $a^* = \arg\max_{a'} \mathbb{E}\left[r|a', \theta\right]$. However for many realistic scenarios (including for matrix completion), sampling from $P(\theta|O_{1:t})$ is not computationally efficient and thus recourse to approximate methods is required to make TS practical.

We propose a computationally-efficient algorithm for solving our problem, which we call *Particle Thompson sampling for matrix factorization (PTS)*. PTS is a combination of particle filtering for online Bayesian parameter estimation and TS in the non-conjugate case when the posterior does not have a closed form. Particle filtering uses a set of weighted samples (particles) to estimate the posterior density. In order to overcome the problem of the huge parameter space, we utilize Rao-Blackwellization and design a suitable Monte Carlo kernel to come up with a computationally and statistically efficient way to update the set of particles as new data arrives in an online fashion. Unlike the prior work [2] which approximates the posterior of the latent item features by a single point estimate, our approach can maintain a much better approximation of the posterior of the latent features by a diverse set of particles. Our results on five different real datasets show a substantial improvement in the cumulative regret vis-a-vis other online methods.

## 2 Probabilistic Matrix Factorization

We first review the probabilistic matrix factorization approach to the low-rank matrix completion problem. In matrix completion, a portion $R^o$ of the $N \times M$ matrix $R = (r_{ij})$ is observed, and the goal is to infer the unobserved entries of $R$. In probabilistic matrix factorization (PMF) [5], $R$ is assumed to be a noisy perturbation of a rank-$K$ matrix $\bar{R} = UV^\top$ where $U_{N \times K}$ and $V_{M \times K}$ are termed the user and item latent features ($K$ is typically small). The full generative model of PMF is

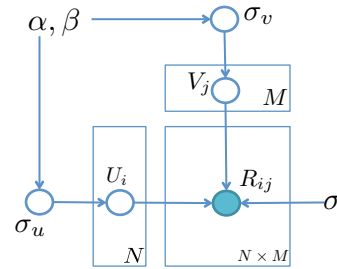

$$\begin{aligned} U_i \text{ i.i.d.} &\sim & \mathcal{N}(0, \sigma_u^2 I_K) \\ V_j \text{ i.i.d.} &\sim & \mathcal{N}(0, \sigma_v^2 I_K) \\ r_{ij}|U, V \text{ i.i.d.} &\sim & \mathcal{N}(U_i^\top V_j, \sigma^2) \end{aligned} \quad (1)$$

Figure 1: Graphical model of probabilistic matrix factorization model

where the variances $(\sigma^2, \sigma_U^2, \sigma_V^2)$ are the parameters of the model. We also consider a full Bayesian treatment where the variances $\sigma_U^2$ and $\sigma_V^2$ are drawn from an inverse Gamma prior (while $\sigma^2$ is held fixed), i.e., $\lambda_U = \sigma_U^{-2} \sim \Gamma(\alpha, \beta)$; $\lambda_V = \sigma_V^{-2} \sim \Gamma(\alpha, \beta)$ (this is a special case of the Bayesian PMF [6] where we only consider isotropic Gaussians)[1]. Given this generative model, from the observed ratings $R^o$, we would like to estimate the parameters $U$ and $V$ which will allow us to "complete" the matrix $R$. PMF is a MAP point-estimate which finds $U, V$ to maximize $\Pr(U, V|R^o, \sigma, \sigma_U, \sigma_V)$ via (stochastic) gradient ascend (alternate least square can also be used [1]). Bayesian PMF [6] attempts to approximate the full posterior $\Pr(U, V|R^o, \sigma, \alpha, \beta)$. The joint posterior of $U$ and $V$ are intractable; however, the structure of the graphical model (Fig. 1) can be exploited to derive an efficient Gibbs sampler.

We now provide the expressions for the conditional probabilities of interest. Supposed that $V$ and $\sigma_U$ are known. Then the vectors $U_i$ are independent for each user $i$. Let $rts(i) = \{j|r_{ij} \in R^o\}$ be the set of items rated by user $i$, observe that the ratings $\{R_{ij}^o|j \in rts(i)\}$ are generated i.i.d. from $U_i$

following a simple conditional linear Gaussian model. Thus, the posterior of $U_i$ has the closed form

$$\Pr(U_i|V, R^o, \sigma, \sigma_U) = \Pr(U_i|V_{rts(i)}, R^o_{i,rts(i)}, \sigma_U, \sigma) = \mathcal{N}(U_i|\mu^u_i, (\Lambda^u_i)^{-1}) \quad (2)$$

$$\text{where } \mu^u_i = \frac{1}{\sigma^2}(\Lambda^u_i)^{-1}\zeta^u_i; \quad \Lambda^u_i = \frac{1}{\sigma^2}\sum_{j \in rts(i)} V_j V_j^\top + \frac{1}{\sigma^2_u}I_K; \quad \zeta^u_i = \sum_{j \in rts(i)} r^o_{ij} V_j. \quad (3)$$

The conditional posterior of $V$, $\Pr(V|U, R^o, \sigma_V, \sigma)$ is similarly factorized into $\prod_{j=1}^M \mathcal{N}(V_j|\mu^v_j, (\Lambda^v_j)^{-1})$ where the mean and precision are similarly defined. The posterior of the precision $\lambda_U = \sigma_U^{-2}$ given $U$ (and simiarly for $\lambda_V$) is obtained from the conjugacy of the Gamma prior and the isotropic Gaussian

$$\Pr(\lambda_U|U, \alpha, \beta) = \Gamma(\lambda_U|\frac{NK}{2} + \alpha, \frac{1}{2}\|U\|^2_F + \beta). \quad (4)$$

Although not required for Bayesian PMF, we give the likelihood expression

$$\Pr(R_{ij} = r|V, R^o, \sigma_U, \sigma) = \mathcal{N}(r|V_j^\top \mu^u_i, \frac{1}{\sigma^2} + V_j^\top \Lambda_{V,i} V_j). \quad (5)$$

The advantage of the Bayesian approach is that uncertainty of the estimate of $U$ and $V$ are available which is crucial for exploration in a bandit setting. However, the bandit setting requires maitaining online estimates of the posterior as the ratings arrive over time which makes it rather awkward for MCMC. In this paper, we instead employ a sequential Monte-Carlo (SMC) method for online Bayesian inference [7, 8]. Similar to the Gibbs sampler [6], we exploit the above closed form updates to design an efficient Rao-Blackwellized particle filter [9] for maintaining the posterior over time.

## 3 Matrix-Factorization Recommendation Bandit

In a typical deployed recommendation system, users and observed ratings (also called *rewards*) arrive over time, and the task of the system is to recommend item for each user so as to maximize the accumulated expected rewards. The bandit setting arises from the fact that the system needs to learn over time what items have the best ratings (for a given user) to recommend, and at the same time sufficiently explore all the items.

We formulate the matrix factorization bandit as follows. We assume that ratings are generated following Eq. (1) with a fixed but unknown latent features $(U^*, V^*)$. At time $t$, the environment chooses user $i_t$ and the system (learning agent) needs to recommend an item $j_t$. The user then rates the recommended item with rating $r_{i_t,j_t} \sim \mathcal{N}(U_{i_t}^{*\top} V_{j_t}^*, \sigma^2)$ and the agent receives this rating as a reward. We abbreviate this as $r^o_t = r_{i_t,j_t}$. The system recommends item $j_t$ using a policy that takes into account the history of the observed ratings prior to time $t$, $r^o_{1:t-1}$, where $r^o_{1:t} = \{(i_k, j_k, r^o_k)\}^t_{k=1}$. The highest expected reward the system can earn at time $t$ is $\max_j U_i^{*\top} V_j^*$, and this is achieved if the optimal item $j^*(i) = \arg\max_j U_i^{*\top} V_j^*$ is recommended. Since $(U^*, V^*)$ are unknown, the optimal item $j^*(i)$ is also not known a priori. The quality of the recommendation system is measured by its *expected cumulative regret*:

$$CR = \mathbb{E}\left[\sum_{t=1}^n [r^o_t - r_{i_t,j^*(i_t)}]\right] = \mathbb{E}\left[\sum_{t=1}^n [r^o_t - \max_j U_{i_t}^{*\top} V_j^*]\right] \quad (6)$$

where the expectation is taken with respect to the choice of the user at time $t$ and also the randomness in the choice of the recommended items by the algorithm.

### 3.1 Particle Thompson Sampling for Matrix Factorization Bandit

While it is difficult to optimize the cumulative regret directly, TS has been shown to work well in practice for contextual linear bandit [3]. To use TS for matrix factorization bandit, the main difficulty is to incrementally update the posterior of the latent features $(U, V)$ which control the reward structure. In this subsection, we describe an efficient Rao-Blackwellized particle filter (RBPF) designed to exploit the specific structure of the probabilistic matrix factorization model. Let $\theta = (\sigma, \alpha, \beta)$ be the control parameters and let posterior at time $t$ be $p_t = \Pr(U, V, \sigma_U, \sigma_V, |r^o_{1:t}, \theta)$. The standard

**Algorithm 1** Particle Thompson Sampling for Matrix Factorization (PTS)
Global control params: $\sigma, \sigma_U, \sigma_V$; for Bayesian version (PTS-B): $\sigma, \alpha, \beta$

1:   $\hat{p}_0 \leftarrow$ InitializeParticles()
2:   $R^o = \emptyset$
3:   **for** $t = 1, 2 \ldots$ **do**
4:       $i \leftarrow$ current user
5:       Sample $d \sim \hat{p}_{t-1}.w$
6:       $\tilde{V} \leftarrow \hat{p}_{t-1}.V^{(d)}$
7:       [If PTS-B] $\tilde{\sigma}_U \leftarrow \hat{p}_{t-1}.\sigma_U^{(d)}$
8:       Sample $\tilde{U}_i \sim \Pr(U_i | \tilde{V}, \tilde{\sigma}_U, \sigma, r_{1:t-1}^o)$            $\triangleright$ sample new $U_i$ due to Rao-Blackwellization
9:       $\hat{j} \leftarrow \arg\max_j \tilde{U}_i^\top \tilde{V}_j$
10:     Recommend $\hat{j}$ for user $i$ and observe rating $r$.
11:     $r_t^o \leftarrow (i, \hat{j}, r)$
12:     $\hat{p}_t \leftarrow$ UpdatePosterior($\hat{p}_{t-1}, r_{1:t}^o$)
13:   **end for**
14:  **procedure** UPDATEPOSTERIOR($\hat{p}, r_{1:t}^o$)
15:                     $\triangleright$ $\hat{p}$ has the structure $(w, \text{particles})$ where $\text{particles}[d] = (U^{(d)}, V^{(d)}, \sigma_U^{(d)}, \sigma_V^{(d)})$.
16:     $(i, j, r) \leftarrow r_t^o$
17:     $\forall d,\ \Lambda_i^{u(d)} \leftarrow \Lambda_i^u(V^{(d)}, r_{1:t-1}^o),\ \zeta_i^{u(d)} \leftarrow \zeta_i^u(V^{(d)}, r_{1:t-1}^o)$       $\triangleright$ see Eq. (3)
18:     $\forall d, w_d \propto \Pr(R_{ij} = r | V^{(d)}, \sigma_U^{(d)}, \sigma, r_{1:t-1}^o)$, see Eq.(5), $\sum w_d = 1$     $\triangleright$ Reweighting; see Eq.(5)
19:     $\forall d,\ i \sim \hat{p}.w; \hat{p}'.\text{particles}[d] \leftarrow \hat{p}.\text{particles}[i];\ \forall d, \hat{p}'.w_d \leftarrow \frac{1}{D}$     $\triangleright$ Resampling
20:     **for** all $d$ **do**                                                            $\triangleright$ Move
21:         $\Lambda_i^{u(d)} \leftarrow \Lambda_i^{u(d)} + \frac{1}{\sigma^2} V_j V_j^\top;\ \zeta_i^{u(d)} \leftarrow \zeta_i^{u(d)} + r V_j$
22:         $\hat{p}'.U_i^{(d)} \sim \Pr(U_i | \hat{p}'.V^{(d)}, \hat{p}'.\sigma_U^{(d)}, \sigma, r_{1:t}^o)$        $\triangleright$ see Eq. (2)
23:         [If PTS-B] Update the norm of $\hat{p}'.U^{(d)}$
24:         $\Lambda_j^{v(d)} \leftarrow \Lambda_j^v(V^{(d)}, r_{1:t}^o),\ \zeta_j^{v(d)} \leftarrow \zeta_i^u(V^{(d)}, r_{1:t}^o)$
25:         $\hat{p}'.V_j^{(d)} \sim \Pr(V_j | \hat{p}'.U^{(d)}, \hat{p}'.\sigma_V^{(d)}, \sigma, r_{1:t}^o)$
26:         [If PTS-B] $\hat{p}'.\sigma_U^{(d)} \sim \Pr(\sigma_U | \hat{p}'.U^{(d)}, \alpha, \beta)$        $\triangleright$ see Eq.(4)
27:     **end for**
28:     return $\hat{p}'$
29:  **end procedure**

particle filter would sample all of the parameters $(U, V, \sigma_U, \sigma_V)$. Unfortunately, in our experiments, degeneracy is highly problematic for such a vanilla particle filter (PF) even when $\sigma_U, \sigma_V$ are assumed known (see Fig. 4(b)). Our RBPF algorithm maintains the posterior distribution $p_t$ as follows. Each of the particle conceptually represents a point-mass at $V, \sigma_U$ ($U$ and $\sigma_V$ are integrated out analytically whenever possible)[2]. Thus, $p_t(V, \sigma_U)$ is approximated by $\hat{p}_t = \frac{1}{D} \sum_{d=1}^{D} \delta_{(V^{(d)}, \sigma_U^{(d)})}$ where $D$ is the number of particles.

Crucially, since the particle filter needs to estimate a set of non-time-vayring parameters, having an effective and efficient MCMC-kernel move $K_t(V', \sigma_U'; V, \sigma_U)$ stationary w.r.t. $p_t$ is essential. Our design of the move kernel $K_t$ are based on two observations. First, we can make use of $U$ and $\sigma_V$ as *auxiliary* variables, effectively sampling $U, \sigma_V | V, \sigma_U \sim p_t(U, \sigma_V | V, \sigma_U)$, and then $V', \sigma_U' | U, \sigma_V \sim p_t(V', \sigma_U' | U, \sigma_V)$. However, this move would be highly inefficient due to the number of variables that need to be sampled at each update. Our second observation is the key to an efficient implementation. Note that latent features for all users except the current user $U_{-i_t}$ are independent of the current observed rating $r_t^o$: $p_t(U_{-i_t} | V, \sigma_U) = p_{t-1}(U_{-i_t} | V, \sigma_U)$, therefore at time $t$ we only have to resample $U_{i_t}$ as there is no need to resample $U_{-i_t}$. Furthermore, it suffices to resample the latent feature of the current item $V_{j_t}$. This leads to an efficient implementation of the RBPF where each particle in fact stores[3] $U, V, \sigma_U, \sigma_V$, where $(U, \sigma_V)$ are auxiliary variables, and for the kernel move $K_t$, we sample $U_{i_t} | V, \sigma_U$ then $V_{j_t}' | U, \sigma_V$ and $\sigma_U' | U, \alpha, \beta$.

The PTS algorithm is given in Algo. 1. At each time $t$, the complexity is $\mathcal{O}(((\hat{N} + \hat{M})K^2 + K^3)D)$ where $\hat{N}$ and $\hat{M}$ are the maximum number of users who have rated the same item and the maximum

number of items rated by the same user, respectively. The dependency on $K^3$ arises from having to invert the precision matrix, but this is not a concern since the rank $K$ is typically small. Line 24 can be replaced by an incremental update with caching: after line 22, we can incrementally update $\Lambda_j^v$ and $\zeta_j^v$ for all item $j$ previously rated by the current user $i$. This reduces the complexity to $\mathcal{O}((\hat{M}K^2 + K^3)D)$, a potentially significant improvement in a real recommendation systems where each user tends to rate a small number of items.

## 4   Analysis

We believe that the regret of PTS can be bounded. However, the existing work on TS and bandits does not provide sufficient tools for proper analysis of our algorithm. In particular, while existing techniques can provide $O(\log T)$ (or $O(\sqrt{T})$ for gap-independent) regret bounds for our problem, these bounds are typically linear in the number of entries of the observation matrix $R$ (or at least linear in the number of users), which is typically very large, compared to $T$. Thus, an ideal regret bound in our setting is the one that has sub-linear dependency (or no dependency at all) on the number of users. A key obstacle of achieving this is that, while the conditional posteriors of $U$ and $V$ are Gaussians, neither their marginal and joint posteriors belong to well behaved classes (e.g., conjugate posteriors, or having closed forms). Thus, novel tools, that can handle generic posteriors, are needed for efficient analysis. Moreover, in the general setting, the correlation between $R^o$ and the latent features $U$ and $V$ are non-linear (see, e.g., [10, 11, 12] for more details). As existing techniques are typically designed for efficiently learning linear regressions, they are not suitable for our problem. Nevertheless, we show how to bound the regret of TS in a very specific case of $n \times m$ rank-1 matrices, and we leave the generalization of these results for future work.

In particular, we analyze the regret of PTS in the setting of Gopalan *et al.* [13]. We model our problem as follows. The *parameter space* is $\Theta_u \times \Theta_v$, where $\Theta_u = \{d, 2d, \ldots, 1\}^{N \times 1}$ and $\Theta_v = \{d, 2d, \ldots, 1\}^{M \times 1}$ are discretizations of the parameter spaces of rank-1 factors $u$ and $v$ for some integer $1/d$. For the sake of theoretical analysis, we assume that PTS can sample from the full posterior. We also assume that $r_{i,j} \sim \mathcal{N}(u_i^* v_j^*, \sigma^2)$ for some $u^* \in \Theta_u$ and $v^* \in \Theta_u$. Note that in this setting, the highest-rated item in expectation is the same for all users. We denote this item by $j^* = \arg\max_{1 \le j \le M} v_j^*$ and assume that it is uniquely optimal, $u_{j^*}^* > u_j^*$ for any $j \ne j^*$. We leverage these properties in our analysis. The *random variable* $X_t$ at time $t$ is a pair of a random rating matrix $R_t = \{r_{i,j}\}_{i=1,j=1}^{N,M}$ and a random row $1 \le i_t \le N$. The *action* $A_t$ at time $t$ is a column $1 \le j_t \le M$. The *observation* is $Y_t = (i_t, r_{i_t, j_t})$. We bound the regret of PTS as follows.

**Theorem 1.** *For any $\delta \in (0,1)$ and $\epsilon \in (0,1)$, there exists $T^*$ such that PTS on $\Theta_u \times \Theta_v$ recommends items $j \ne j^*$ in $T \ge T^*$ steps at most $(2M \frac{1+\epsilon}{1-\epsilon} \frac{\sigma^2}{d^4} \log T + B)$ times with probability of at least $1 - \delta$, where $B$ is a constant independent of $T$.*

*Proof.* By Theorem 1 of Gopalan *et al.* [13], the number of recommendations $j \ne j^*$ is bounded by $C(\log T) + B$, where $B$ is a constant independent of $T$. Now we bound $C(\log T)$ by counting the number of times that PTS selects models that cannot be distinguished from $(u^*, v^*)$ after observing $Y_t$ under the optimal action $j^*$. Let:

$$\Theta_j = \left\{ (u,v) \in \Theta_u \times \Theta_v : \forall i : u_i v_{j^*} = u_i^* v_{j^*}^*, \ v_j \ge \max_{k \ne j} v_k \right\}$$

be the set of such models where action $j$ is optimal. Suppose that our algorithm chooses model $(u,v) \in \Theta_j$. Then the KL divergence between the distributions of ratings $r_{i,j}$ under models $(u,v)$ and $(u^*, v^*)$ is bounded from below as:

$$D_{\mathrm{KL}}(u_i v_j \,\|\, u_i^* v_j^*) = \frac{(u_i v_j - u_i^* v_j^*)^2}{2\sigma^2} \ge \frac{d^4}{2\sigma^2}.$$

for any $i$. The last inequality follows from the fact that $u_i v_j \ge u_i v_{j^*} = u_i^* v_{j^*}^* > u_i^* v_j^*$, because $j^*$ is uniquely optimal in $(u^*, v^*)$. We know that $\left| u_i v_j - u_i^* v_j^* \right| \ge d^2$ because the granularity of our discretization is $d$. Let $i_1, \ldots, i_n$ be any $n$ row indices. Then the KL divergence between the distributions of ratings in positions $(i_1, j), \ldots, (i_n, j)$ under models $(u,v)$ and $(u^*, v^*)$ is $\sum_{t=1}^n D_{\mathrm{KL}}(u_{i_t} v_j \,\|\, u_{i_t}^* v_j^*) \ge n \frac{d^4}{2\sigma^2}$. By Theorem 1 of Gopalan *et al.* [13], the models $(u,v) \in \Theta_j$ are unlikely to be chosen by PTS in $T$ steps when $\sum_{t=1}^n D_{\mathrm{KL}}(u_{i_t} v_j \,\|\, u_{i_t}^* v_j^*) \ge \log T$. This happens after at most $n \ge 2 \frac{1+\epsilon}{1-\epsilon} \frac{\sigma^2}{d^4} \log T$ selections of $(u,v) \in \Theta_j$. Now we apply the same argument to all $\Theta_j$, $M-1$ in total, and sum up the corresponding regrets. $\blacksquare$

**Remarks**: Note that Theorem 1 implies at $O(2M\frac{1+\epsilon}{1-\epsilon}\frac{\sigma^2}{d^4}\log T)$ regret bound that holds with high probability. Here, $d^2$ plays the role of a gap $\Delta$, the smallest possible difference between the expected ratings of item $j \neq j^*$ in any row $i$. In this sense, our result is $O((1/\Delta^2)\log T)$ and is of a similar magnitude as the results in Gopalan *et al.* [13]. While we restrict $u^*, v^* \in (0,1]^{K\times 1}$ in the proof, this does not affect the algorithm. In fact, the proof only focuses on high probability events where the samples from the posterior are concentrated around the true parameters, and thus, are within $(0,1]^{K\times 1}$ as well. Extending our proof to the general setting is not trivial. In particular, moving from discretized parameters to continuous space introduces the abovementioned ill behaved posteriors. While increasing the value of K will violate the fact that the best item will be the same for all users, which allowed us to eliminate $N$ from the regret bound.

## 5 Experiments and Results

The goal of our experimental evaluation is twofold: (i) evaluate the PTS algorithm for making online recommendations with respect to various baseline algorithms on several real-world datasets and (ii) understand the qualitative performance and intuition of PTS.

### 5.1 Dataset description

We use a synthetic dataset and five real world datasets to evaluate our approach. The synthetic dataset is generated as follows - At first we generate the user and item latent features ($U$ and $V$) of rank $K$ by drawing from a Gaussian distribution $\mathcal{N}(0, \sigma_u^2)$ and $\mathcal{N}(0, \sigma_v^2)$ respectively. The true rating matrix is then $R^* = UV^T$. We generate the observed rating matrix $R$ from $R^*$ by adding Gaussian noise $\mathcal{N}(0, \sigma^2)$ to the true ratings. We use five real world datasets as follows: Movielens 100k, Movielens 1M, Yahoo Music[4], Book crossing[5] and EachMovie as shown in Table 1.

|  | Movielens 100k | Movielens 1M | Yahoo Music | Book crossing | EachMovie |
|---|---|---|---|---|---|
| # users | 943 | 6040 | 15400 | 6841 | 36656 |
| # items | 1682 | 3900 | 1000 | 5644 | 1621 |
| # ratings | 100k | 1M | 311,704 | 90k | 2.58M |

Table 1: Characteristics of the datasets used in our study

### 5.2 Baseline measures

There are no current approaches available that simultaneously learn both the user and item factors by sampling from the posterior in a bandit setting. From the currently available algorithms, we choose two kinds of baseline methods - one that sequentially updates the the posterior of the user features only while fixing the item features to a point estimate (ICF) and another that updates the MAP estimates of user and item features via stochastic gradient descent (SGD-Eps). A key challenge in online algorithms is unbiased offline evaluation. One problem in the offline setting is the partial information available about user feedback, i.e., we only have information about the items that the user rated. In our experiment, we restrict the recommendation space of all the algorithms to recommend among the items that the user rated in the entire dataset which makes it possible to empirically measure regret at every interaction. The baseline measures are as follows:

1) **Random** : At each iteration, we recommend a random movie to the user.

2) **Most Popular** : At each iteration, we recommend the most popular movie restricted to the movies rated by the user on the dataset. Note that this is an unrealistically optimistic baseline for an online algorithm as it is not possible to know the global popularity of the items beforehand.

3) **ICF**: The ICF algorithm [2] proceeds by first estimating the user and item latent factors ($U$ and $V$) on a initial training period and then for every interaction thereafter only updates the user features ($U$) assuming the item features ($V$) as fixed. We run two scenarios for the ICF algorithm one in which we use 20% (icf-20) and 50% (icf-50) of the data as the training period respectively. During this period of training, we randomly recommend a movie to the user to compute the regret. We use the PMF implementation by [5] for estimating the $U$ and $V$.

4) **SGD-Eps**: We learn the latent factors using an online variant of the PMF algorithm [5]. We use the stochastic gradient descent to update the latent factors with a mini-batch size of 50. In order to make a recommendation, we use the $\epsilon$-greedy strategy and recommend the highest $U_iV^T$ with a probability $\epsilon$ and make a random recommendations otherwise. ($\epsilon$ is set as 0.95 in our experiments.)

## 5.3 Results on Synthetic Dataset

We generated the synthetic dataset as mentioned earlier and run the PTS algorithm with 100 particles for recommendations. We simulate the setting as mentioned in Section 3 and assume that at time t, a random user $i_t$ arrives and the system recommends an item $j_t$. The user rates the recommended item $r_{i_t,j_t}$ and we evaluate the performance of the model by computing the expected cumulative regret defined in Eq(6). Fig. 2 shows the cumulative regret of the algorithm on the synthetic data averaged over 100 runs using different size of the matrix and latent features $K$. The cumulative regret increases sub-linearly with the number of interactions and this gives us confidence that our approach works well on the synthetic dataset.

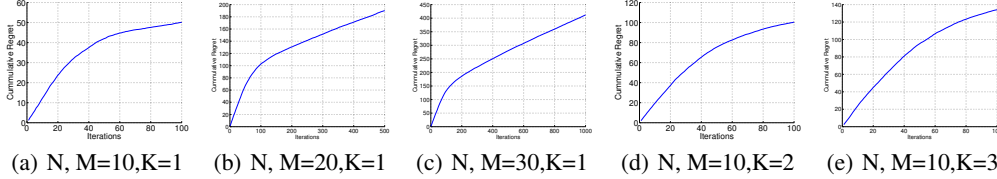

(a) N, M=10,K=1    (b) N, M=20,K=1    (c) N, M=30,K=1    (d) N, M=10,K=2    (e) N, M=10,K=3

Figure 2: Cumulative regret on different sizes of the synthetic data and $K$ averaged over 100 runs.

## 5.4 Results on Real Datasets

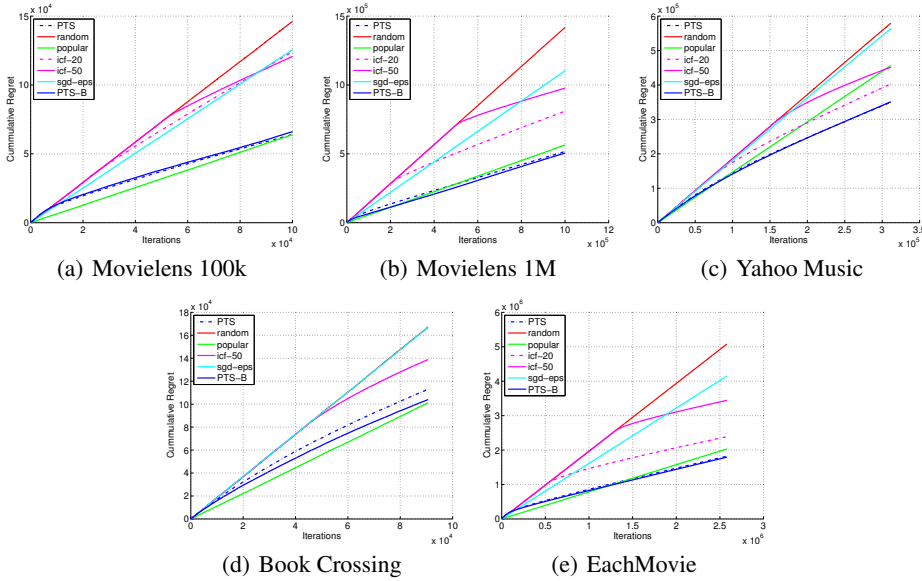

(a) Movielens 100k          (b) Movielens 1M          (c) Yahoo Music

(d) Book Crossing          (e) EachMovie

Figure 3: Comparison with baseline methods on five datasets.

Next, we evaluate our algorithms on five real datasets and compare them to the various baseline algorithms. We subtract the mean ratings from the data to centre it at zero. To simulate an extreme cold-start scenario we start from an empty set of user and rating. We then iterate over the datasets and assume that a random user $i_t$ has arrived at time t and the system recommends an item $j_t$ constrained to the items rated by this user in the dataset. We use $K = 2$ for all the algorithms and use 30 particles for our approach. For PTS we set the value of $\sigma^2 = 0.5$ and $\sigma_u^2 = 1, \sigma_v^2 = 1$. For PTS-B (Bayesian version, see Algo. 1 for more details), we set $\sigma^2 = 0.5$ and the initial shape parameters of the Gamma distribution as $\alpha = 2$ and $\beta = 0.5$. For both ICF-20 and ICF-50, we set $\sigma^2 = 0.5$ and $\sigma_u^2 = 1$. Fig. 3 shows the cumulative regret of all the algorithms on the five datasets[6]. Our approach performs significantly better as compared to the baseline algorithms on this diverse set of datasets. PTS-B with no parameter tuning performs slightly better than PTS and achieves the best regret. It is important to note that both PTS and PTS-B performs comparable to or even better than the "most popular" baseline despite not knowing the global popularity in advance. Note that ICF is very sensitive to the length of the initial training period; it is not clear how to set this apriori.

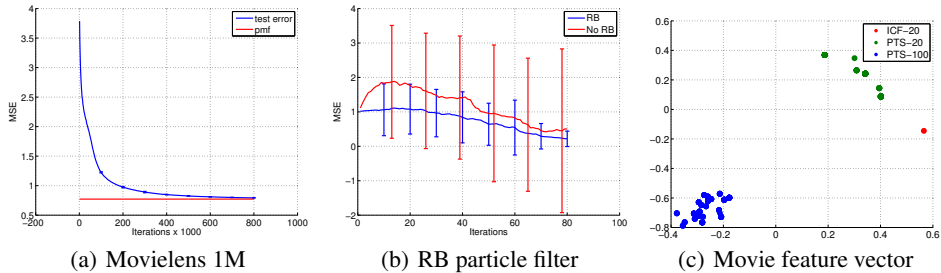

|  (a) Movielens 1M | (b) RB particle filter | (c) Movie feature vector |

Figure 4: a) shows MSE on movielens 1M dataset, the red line is the MSE using the PMF algorithm b) shows performance of a RBPF (blue line) as compared to vanilla PF (red line) on a synthetic dataset N,M=10 and c) shows movie feature vectors for a movie with 384 ratings, the red dot is the feature vector from the ICF-20 algorithm (using 73 ratings). PTS-20 is the feature vector at 20% of the data (green dots) and PTS-100 at 100% (blue dots).

We also evaluate the performance of our model in an offline setting as follows: We divide the datasets into training and test set and iterate over the training data triplets $(i_t, j_t, r_t)$ by pretending that $j_t$ is the movie recommended by our approach and update the latent factors according to RBPF. We compute the recovered matrix $\hat{R}$ as the average prediction $UV^T$ from the particles at each time step and compute the mean squared error (MSE) on the test dataset at each iteration. Unlike the batch method such as PMF which takes multiple passes over the data, our method was designed to have bounded update complexity at each iteration. We ran the algorithm using 80% data for training and the rest for testing and computed the MSE by averaging the results over 5 runs. Fig. 4(a) shows the average MSE on the movielens 1M dataset. Our MSE (0.7925) is comparable to the PMF MSE (0.7718) as shown by the red line. This demonstrates that the RBPF is performing reasonably well for matrix factorization. In addition, Fig. 4(b) shows that on the synthetic dataset, the vanilla PF suffers from degeneration as seen by the high variance. To understand the intuition why fixing the latent item features $V$ as done in the ICF does not work, we perform an experiment as follows: We run the ICF algorithm on the movielens 100k dataset in which we use 20% of the data for training. At this point the ICF algorithm fixes the item features $V$ and only updates the user features $U$. Next, we run our algorithm and obtain the latent features. We examined the features for one selected movie from the particles at two time intervals - one when the ICF algorithm fixes them at 20% and another one in the end as shown in the Fig. 4(c). It shows that movie features have evolved into a different location and hence fixing them early is not a good idea.

## 6 Related Work

Probabilistic matrix completion in a bandit setting setting was introduced in the previous work by Zhao *et al.* [2]. The ICF algorithm in [2] approximates the posterior of the latent item features by a single point estimate. Several other bandit algorithms for recommendations have been proposed. Valko *et al.* [14] proposed a bandit algorithm for content-based recommendations. In this approach, the features of the items are extracted from a similarity graph over the items, which is known in advance. The preferences of each user for the features are learned independently by regressing the ratings of the items from their features. The key difference in our approach is that we also learn the features of the items. In other words, we learn both the user and item factors, $U$ and $V$, while [14] learn only $U$. Kocak *et al.* [15] combine the spectral bandit algorithm in [14] with TS. Gentile *et al.* [16] propose a bandit algorithm for recommendations that clusters users in an online fashion based on the similarity of their preferences. The preferences are learned by regressing the ratings of the items from their features. The features of the items are the input of the learning algorithm and they only learn $U$. Maillard *et al.* [17] study a bandit problem where the arms are partitioned into unknown clusters unlike our work which is more general.

## 7 Conclusion

We have proposed an efficient method for carrying out matrix factorization ($M \approx UV^T$) in a bandit setting. The key novelty of our approach is the combined use of Rao-Blackwellized particle filtering and Thompson sampling (PTS) in matrix factorization recommendation. This allows us to simultaneously update the posterior probability of $U$ and $V$ in an online manner while minimizing the cumulative regret. The state of the art, till now, was to either use point estimates of $U$ and $V$ or use a point estimate of one of the factor (e.g., $U$) and update the posterior probability of the other ($V$). PTS results in substantially better performance on a wide variety of real world data sets.

## Footnotes

[1][6] considers the full covariance structure, but they also noted that isotropic Gaussians are effective enough.

[2]When there are fewer users than items, a similar strategy can be derived to integrate out $U$ and $\sigma_V$ instead.

[3]This is not inconsistent with our previous statement that conceptually a particle represents only a point-mass distribution $\delta_{V, \sigma_U}$.

[4]http://webscope.sandbox.yahoo.com/

[5]http://www.bookcrossing.com

[6]ICF-20 fails to run on the Bookcrossing dataset as the 20% data is too sparse for the PMF implementation.

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
