[Reviews · NeurIPS 2015]

Submitted by Assigned_Reviewer_1

I believe the authors have misunderstood my request for comparison with other techniques.

My point is that PMF is a weak benchmark in the sense that many other collaborative filtering methods known to be more accurate than PMF can also be trained in an online fashion using SGD.

I would recommend comparing against stronger benchmarks like those discussed in the provided reference or the papers that cite it.
Summary: This paper brings together several existing ideas (bandit collaborative filtering, Bayesian probabilistic matrix factorization, and Thompson sampling) to make an apparently useful contribution to the realm of online collaborative filtering. I would like to see a more comprehensive empirical comparison with existing collaborative filtering procedures trained using SGD over observations, since any of these procedures could be employed in an online recommendation setting (see, e.g., most of the procedures in Koren et al. MATRIX FACTORIZATION TECHNIQUES FOR RECOMMENDER SYSTEMS, many of which are more accurate than PMF). The value of the theory presented is unclear, as the assumptions seem quite strong and unrealistic.

Submitted by Assigned_Reviewer_2

The authors propose an online matrix completion algorithm that employs particle filters to represent the posterior distribution over user/item factors. Thompson sampling is employed to recommend items according to this approximated posterior. Particle are then resampled, but only the vectors of item and user corresponding to the most recently received rating.

A theoretical bound on the regret is provided, and the method is analyzed on synthetic data, and compared against various baseline strategies on 5 real datasets. The method outperforms all but one baseline on all the datasets (and the best baseline is considered unrealistic but see question below). The value of this method hinges on the assumption that uncertainty of *both* user and item factors is necessary to make reasonable recommendations in the cold-start scenario, which is illustrated in Figure 4c.

Overall, this paper may be a solid contribution but I have some concerns that need to be resolved:

1. In 4/5 of the real datasets, the proposed method is either comparable or

worse than the strategy of picking the most popular item (restricted to those recommended for that user). This is labelled as an 'unrealistically optimistic baseline' as it is not possible to know global popularity beforehand. However, it certainly seems possible to estimate global popularity on a training set (akin to how ICF estimates item factors on a training set) and evaluate on a test set. This also seems very possible in a real-world setting (collect information on global popularity before deploying your recommendation system). I think a comparison with such a strategy is warranted.

2. How were K=2 and 30 particles chosen? How does the algorithm scale and perform for larger K? Do you need many more particles to represent posteriors over larger-dimensional objects for these datasets?

3. The authors acknowledge that ICF performance is sensitive to the amount of training data used. A plot showing how ICF performs against PTS/PTS-B as a function of training proportion would be helpful here.

Some other minor points:

1. It's mentioned that Thompson sampling is used to recommend the item, but in Algorithm 1, the item chosen to be recommended is simply \argmax_j \tilde{U}_i^T\tilde{V}_j. Is the ``sampling'' part simply coming from the sampling of \tilde{U} and \tilde{V} using the particles?

2. The method performs slightly worse in an offline setting than standard PMF (0.7925 vs. 0.7718) - is there an explanation for why?

3. The following citable work (obtained via google search of 'matrix completion bandit problem') seems very relevant, yet is not cited:

Linear Bandits, Matrix Completion, and Recommendation Systems. Madeleine Udell and Reza Takapoui. NIPS Workshop on Large Scale Matrix Analysis and Inference, 2014.

Summary: The key contribution is a novel online collaborative filtering algorithm that trades off exploration and exploitation by using particle filtering to represent uncertainty in both user and item factors. The paper is clear and well-written, but there are some limitations/questions around (1) how the model is evaluated, and (2) how this algorithm scales for larger numbers of latent factors.

Submitted by Assigned_Reviewer_3

This paper proposes a new algorithm for probabilistic matrix factorization.

The idea is novel and the contribution is significant.
Summary: The authors proposed a probabilistic matrix factorization algorithm for online recommendation.

The algorithms makes sense and the experiments show promising results.

Submitted by Assigned_Reviewer_4

In this paper, the authors studies the problem of online matrix factorization that can find the most relevant items with exploring cold-start items. The proposed approach is named Particle Thompson sampling for matrix factorization, which can be seen as the general Thompson sampling augmented with online Bayesian probabilistic matrix factorization with Rao-Blackwellized particle filter.

The authors formulate the online matrix factorization problem as a matrix factorization bandit, and introduce the expected cumulative regret, which can be considered as the expectation with respect to the choice of a user at time 5 and also randomness in the choice of the recommended items by the algorithm. Further, the authors proposed an efficient Rao-Blackwellized particle filter (RBPF) for exploit the matrix factorization problem, and show that the regret of the RBPF is bounded. Experiments on five real-world datasets show that the proposed algorithms perform significantly better than a number of baselines.

Quality: the quality of the paper is good as reflected by the theoretical and empirical results presented in the paper.

Clarity: the paper is clearly written.

Originality: the studied problem in this paper is not new, but I think the contribution in terms of the proposed method is new and interesting.

Significance: Empirical results across five real-world datasets demonstrate that the results of this proposed method is significant.
Summary: This paper studies the problem of online matrix factorization that learns to find the most relevant items with exploring cold-start items. The proposed approach is named Particle Thompson sampling for matrix factorization, which can be seen as the general Thompson sampling augmented with online Bayesian probabilistic matrix factorization with Rao-Blackwellized particle filter.

Author Feedback
Author rebuttal: We thank all of the reviewers for their thoughtful comments. Below we respond to specific concerns raised by Reviewers 1 and 6.

Reviewer1

1) "estimate global popularity on a training set"
We performed a comparison with the popular strategy on MovieLens100k dataset with a setup similar to ICF as suggested by Reviewer1. The performance of this strategy is much worse than our PTS and the original oracle popularity baseline.
Intuitively, incremental estimate of popularity does not help when a new item arrives.
See
https://www.dropbox.com/s/8398wp9mlomcu1m/movielens100Knewpoprange.pdf?dl=0

2) "How does the algorithm scale and perform for larger K? Do you need many more particles?"
We ran PTS with K=(3, 5, 7, 9) while keeping the number of particles fixed at 100.
Overall, the performance is roughly the same with these larger Ks.
See https://www.dropbox.com/s/m2e89lpxxt37nqh/movielens100KdiffK.pdf?dl=0
The Rao-Blackwellized Particle Filter works well even with this reasonably small number of particles (100). Note that this is not the case with the
standard PF as seen in Fig. 4(b), therefore we believe that Rao-Blackwellization is the key to keeping the number of particles small. Furthermore, unlike MCMC, the PF can be easily parallelizable.

3) "how ICF performs against PTS/PTSĀ­B as a function of training proportion"
We provide additional results for ICF using training percentage 20,30,40 and 50 -- all variants perform significantly worse than
our PTS. All and observe that the results do not get better varying the training percentage. See
https://www.dropbox.com/s/gu3rpfc0nj0qyzr/movielens100KdiffICF.pdf?dl=0

4) " Is the ``sampling'' part simply coming from the sampling of \tilde{U} and \tilde{V} using the particles?"
\tilde{V} is sampled from the particles. However as the result of Rao-Blackwellization,
\tilde{U_i} is sampled from the true posterior given \tilde{V}, see line 8 in Alg. 1.

5) "The method performs slightly worse in an offline setting than standard PMF"
The PMF algorithm is a batch algorithm and takes several passes (in this case 50) over the entire data.
Our PTS algorithm is online and passes over each rating once. Also, it is known that algorithms designed for cumulative regret minimization cannot perform well in terms of minimizing the estimation error (i.e., simple regret) (see, e.g., Bubek, Munos, and Stolz (2009)). Since PTS is designed to minimize the former, achieving the best accuracy in predicting the rating is not our goal.

6) " work seems very relevant, yet is not cited"
Thanks for the pointer and we will cite the work, although what we can find -- http://web.stanford.edu/~takapoui/bmc.pdf --
is a one-page abstract.

Reviewer 6
"I would like to see a more comprehensive empirical comparison with existing collaborative filtering procedures trained using SGD over observations .."
Thank you for the useful suggestion. We just want to make two small remarks: (a) One of our baselines (SGD-Eps) is using SGD over observations, and is significantly worse than PTS. (b) It is not possible to perform Thompson sampling if one uses SGD as it only provides a point estimate.

"The value of the theory presented is unclear, as the assumptions seem quite
strong and unrealistic."
The main contribution of the paper is to demonstrate the practical efficiency of PTS.
However, we conjecture that the algorithm also enjoys nice theoretical guarantees as well. Unfortunately, as we have also discussed in Section 4, current theoretical tools are not suitable for devising such guarantees in generic settings. Nevertheless, we demonstrate how one can still achieve logarithmic regret bounds in some special cases. We hope that by doing so, we will bring the appetite to the (theoretical) community to focus more to this problem.